# Index Properties, Hydraulic Conductivity and Contaminant-Compatibility of CMC-Treated Sodium Activated Calcium Bentonite

**DOI:** 10.3390/ijerph17061863

**Published:** 2020-03-13

**Authors:** Ri-Dong Fan, Krishna R. Reddy, Yu-Ling Yang, Yan-Jun Du

**Affiliations:** 1School of Materials Science and Engineering, Southeast University, Nanjing 210096, China; fanrd@seu.edu.cn; 2Jiangsu Key Laboratory of Urban Underground Engineering & Environmental Safety, Institute of Geotechnical Engineering, Southeast University, Nanjing 210096, China; ylyang@seu.edu.cn; 3Department of Civil & Materials Engineering, University of Illinois at Chicago, Chicago, IL 60607, USA; kreddy@uic.edu

**Keywords:** bentonite, carboxymethyl cellulose, chemical compatibility, heavy metal, landfill leachate

## Abstract

A typical sodium activated calcium bentonite (SACaB) was treated with carboxymethyl cellulose (CMC) polymer, called CMC-treated SACaB (CMC-SACaB), and it was investigated for its hydraulic conductivity and enhanced chemical compatibility. Index property and hydraulic conductivity tests were conducted on CMC-SACaB and SACaB with deionized water (DIW), heavy metals-laden water, and actual landfill leachate. Lead-zinc mixed (Pb-Zn) solution and hexavalent chromium (Cr(VI)) solution were selected as target heavy metals-laden water, and calcium (Ca) solution was tested for comparison purposes. The hydraulic conductivity (kMFL) was determined via the modified fluid loss (MFL) test. Liquid limit and swell index in DIW, heavy metal-laden water, and Ca solution increased with increasing CMC content. CMC treatment effectively decreased the kMFL of SACaB when exposed to Pb-Zn solutions with a metal concentration of 1 to 20 mmol/L and landfill leachate. An insignificant change in kMFL of CMC-SACaB occurred with exposure to Pb-Zn solutions with metal concentrations of 1 to 10 mmol/L, Cr(VI) and Ca solutions with metal concentration of 1 to 20 mmol/L, and landfill leachate. A slight increase in kMFL of CMC-SACaB was observed when Pb-Zn concentration increased to 20 mmol/L, and such an increment was more noticeable when the CMC content was lower than 10%. In the DIW, the measured kMFL values of CMC-SACaB and SACaB with a given range of void ratio were consistent with those obtained from the flexible-wall permeameter test.

## 1. Introduction

Engineered barriers are extensively used in the containment of groundwater seepage and landfill leachate for the purpose of risk control of contaminated sites and landfill sites [1,2,3]. High quality sodium bentonite (NaB) is one of the most common materials used in the engineered barriers, such as soil-bentonite slurry-trench cut-off walls, geosynthetic clay liners (GCL), and compacted soil-bentonite liners, because of its relatively higher swell potential and lower hydraulic conductivity. However, NaB might not be available in some countries, such as China and India, whereas calcium bentonite (CaB) and sodium activated CaB (SACaB) are abundant and they can be used as an alternative to make up the engineered barriers [4,5]. GMZ bentonite, high-swellabel NaB is particularly applied to high-level radioactive waste repository in China [6,7].

CaB and SACaB show relatively lower swell potential when compared with NaB due to the lower amount of exchangeable sodium ions. For example, the swell index of standard Wyoming bentonite MX-80, a typical NaB, ranges from 24 to 28 mL/2 g; whereas the swell index of CaB commonly ranges as low as 3 to 19 mL/2 g [8,9,10,11]. On the other hand, chemical compatibility is an important consideration for all engineered barriers in geoenvironmental applications [5,12]. The chemical compatibility is generally defined as the degree of change in engineering properties of the barrier due to its exposure to contaminant. The hydraulic conductivity in chemical solutions is one of the most important parameters that is considered in the design of engineered barriers. The hydraulic conductivity is dependent on the properties of both engineered barrier materials and permeating chemical solutions. The hydraulic conductivity of engineered barriers that use either CaB, SACaB, or NaB tends to increase in aggressive chemical solutions, such as salt solutions, landfill leachates, low-level radioactive waste leachates, and coal combustion product leachates with relatively high concentrations of multivalent cations [11,13,14]. Previous studies showed that concentrated salt solutions (e.g., CaCl_2_, Cu(NO_3_)_2_ and Pb(NO_3_)_2_ at metal concentrations of 100 mmol/L) could result in a considerable increase in hydraulic conductivity (e.g., 4- to 100- fold) of various bentonites, GCLs, and sand/Na-bentonite vertical cutoff wall backfills [5,12,15,16]. The reason is attributed to the compression of the diffuse double layer (DDL) of the bentonite particles.

Several innovative efforts have been made to enhance the chemical compatibility of Na-bentonites by using inorganic phosphate, e.g., sodium hexametaphosphate-amended bentonite, and polymeric materials, e.g., bentonite-polymer nanocomposite (BPN), multiswellable bentonite (MSB), and carboxymethyl cellulose (CMC)-treated NaB bentonite (HYPER clay) [11,17,18]. The results indicated that the hydraulic conductivity of the treated-based GCL bentonites could decrease by approximately ten to ten thousand times when compared with that of the untreated bentonites when permeated with inorganic solutions, including KCl, NaCl, CaCl_2_, NaOH, and HNO_3_ [17,18,19,20]. The hydraulic conductivity of soil-bentonite slurry-trench cutoff wall backfills could decrease by approximately one to two orders of magnitude when conventional NaB was replaced with bentonite-polymer nanocomposite when permeated with tap water and CaCl_2_ solution with a concentration of 50 mmol/L [17]. In addition, the drilling fluid that was made of HYPER clay performed better in terms a lower filtrate volume after 30 min. in seawater as compared to the untreated drilling fluid that was made of NaB [21]. Nevertheless, only limited studies have investigated the chemical compatibility of NaB, SACaB and bentonite-based engineered barriers using real contaminants, such as heavy metal solutions, low-level radioactive waste leachates, and coal combustion product leachates [5,11,13,14]. It should be mentioned that knowledge of the effect of bentonite modifier dosage on the chemical compatibility is essential in the application of treated bentonites in real projects. However, the method for determining the optimized modifier dosage for a certain bentonite has not been well addressed. A limited study reported by Janssen et al. [9] investigated the impact of CaCl_2_ solution on the swell pressure and hydraulic conductivity of CaB and SACaB that were treated with 12% and 16% CMC. The effect of CMC content (0 to 16%) on the swell index of bentonite was investigated under CaCl_2_ solution exposure. Their results indicated that the swell index noticeably increased with increasing CMC content when the Ca concentration was lower than 10 mmol/L.

The research motivation for this study is summarized as follows: (1) how SACaB can be treated by using a polymer material to improve its chemical compatibility when exposed to ground water with heavy metal contaminants or landfill leachate; and, (2) how to optimize the polymer content in the SACaB for an effective containment barrier with enhanced chemical compatibility. With these questions in mind, carboxymethyl cellulose (CMC) is used to treat SACaB and investigate the index properties and chemical compatibility of CMC-treated SACaB (CMC-SACaB) and compare these results with those of the CMC-treated NaB (i.e., HYPER clays). The swell index and hydraulic conductivity are selected as the target properties for evaluating the improvement that results from the CMC treatment. The CMC content in bentonite is optimized by exploring the effect that CMC content has on the hydraulic conductivity of CMC-treated bentonite that is exposed to aqueous solution with real contaminants. CMC is used to treat the SACaB, because: (1) CMC, as one kind of hydrophilic anionic polymer, shows high potential to reduce the hydraulic conductivity of Na-bentonite when it is exposed to KCl and CaCl_2_ solutions [19]; (2) standardized production of CMC-treated bentonite is practicable as CMC has been used as a standard thickening reagent for bentonite-water slurry in drilling engineering [22]; and (3) the main chemical interaction between bentonite and CMC involves the sorption of CMC on bentonite, which results in a relatively simple method for CMC-treated bentonite preparation [23].

Index properties, including liquid limit, plastic limit, specific gravity, and swell index, are fundamental properties of bentonites [10]. Moreover, liquid limit and swell index can be considered as a basis for the initial assessment of hydraulic conductivity and contaminant-compatibility of clayey soils being potentially used as engineered barrier materials. For instance, a bentonite with relatively higher values of liquid limit or swell index generally possesses lower hydraulic conductivity at a given void ratio. Hence, liquid limit, swell index, and void ratio are used to establish models for predicting hydraulic conductivity of fine-grained soils [24]. Liquid limit ratio, which is defined as the ratio of liquid limit of contaminated soil to that of clean soil prepared with deionized water, can be used to assess the change of hydraulic conductivity of clayey soils in aggressive chemical solutions [5,25].

A modified fluid loss (MFL) test is used in this study to evaluate the hydraulic conductivity of bentonite and assess the potential optimized CMC content in the SACaB. The MFL test, which was originally from the American Petroleum Institute filtration test [26], was developed to evaluate the hydraulic conductivity of bentonites by Chung and Daniel [27]. In recent years, the MFL test has been considered as a quick method for investigating the chemical compatibility of bentonite as compared with the conventional permeability test that was permeated with inorganic salt solutions (e.g., NaCl, CaCl_2_), landfill leachate, and acid mining drainage [28,29,30]. Evaluating the hydraulic conductivity using the MFL test has the following advantages over that directly measured using a flexible wall permeameter (FWP): (1) the evaluation of hydraulic conductivity via the MFL test can be done within only 24 h, but it would likely take weeks to months to accomplish for a FWP test [28], and (2) the MFL test generally yields a comparable or conservative hydraulic conductivity as compared with that measured via the FWP test when the bentonites are permeated with distilled-deionized water or chemical solutions, though very limited data reported in the literature showed that the MFL test could result in a lower hydraulic conductivity [27,28,29].

## 2. Materials and Methods

### 2.1. SACaB

The MUFENF mineral processing plant in Zhenjiang City, China provided the SACaB that was used here. Table 1 shows its physico-chemical properties, as determined per ASTM standards. The resulting swell index is 16.5 mL/2 g, which is approximately one-half lower than that of the typical commercial NaB used for engineered barriers in practice (32 to 35 mL/2 g).

### 2.2. CMC-SACaB Preparation

The CMC was provided by ACROS Organics (Pittsburgh, PA, USA). It is white in color with an average molecular weight of 700000, and the degree of substitution (DS) is 0.9, which indicates its relatively high dissolution potential in water. The solubility of CMC in water could be as high as 2%, and the recommended temperature for dissolving CMC in the water is 60 °C [18]. The CMC content in bentonite (CCB) was set at 2%, 6%, 10%, and 14%. The CMC-SACaB sample is designated as CMCi in order to denote a bentonite with CMC content of i%.

The method for CMC-SACaB preparation is based on Di Emidio et al. [18]. A predetermined mass of CMC was mixed with a predetermined volume of deionized water (DIW) at a temperature of 60 °C for 30 min. using a paddle mixer. The CMC-DIW mixture was allowed to remain for 3 min. at 60 °C; and then, the pH of the mixture was measured at 60 °C using a Model 720 A pH meter that was manufactured by Cole-Parmer Instrument Company, Vernon Hills, IL. After that, a predetermined mass of SACaB was mixed with the CMC-DIW mixture for 3 h using a paddle mixer. A water bath was used to control the temperature of 60 °C during the mixing. The time for mixing the SACaB with the CMC-DIW mixture in this study is much longer than that applied by Di Emidio et al. [18] (30 min.) for the sake of a sufficient chemical interaction between the SACaB and CMC. The CMC-SACaB slurry was then dried in an oven at 105 °C for 24 h. Finally, the dried CMC-treated bentonite was sieved through the No. 200 (75 µm) sieve and that passed through the sieve was used in this study.

### 2.3. Chemical Solutions

A mixture of lead and zinc (Pb-Zn) and hexavalent chromium (Cr(VI)) were selected as the simulated heavy-metal contaminants, because these are commonly found in contaminated groundwater. DIW was used as a baseline to evaluate the chemical compatibility of CMC-SACaB and SACaB. Benson et al. [39] also suggested DIW as a baseline to evaluate the chemical compatibility of GCLs. In addition, calcium (Ca) solution was also used for testing for comparison with previously published studies. The Pb(NO_3_)_2_-Zn(NO_3_)_2_, K_2_CrO_4_, and CaCl_2_ were selected as chemical sources for preparing Pb-Zn, Cr(VI), and Ca solutions, respectively, and they were prepared by dissolving a predetermined mass of chemical powder (ACS reagent) in DIW with a predetermined volume to yield target metal concentration. The ratio of molar concentration of Pb to that of Zn in the Pb-Zn solution was controlled as 1.0. The DIW (pH = 6.90; electrical conductivity, *EC* = 4 µS/cm) is classified as Type IV based on ASTM D1193 [40]. Concentration of aqueous species and ionic strength were calculated by using Visual MINTEQ (Ver 3.1.) to understand the chemical speciation of metals in the aqueous solution. Figure 1 shows the comparison between the concentration of aqueous species and total metal concentration. The results indicate that metals in Ca, Pb-Zn, and Cr(VI) solutions are in the form of hydrated ions, i.e., Ca^2+^, Pb^2+^ and PbNO^3+^, Pb^2+^ and PbNO_3_^+^, CrO_4_^-^, and K^+^, respectively. Table 2 presents the ionic strength, absolute viscosity (*μ*), unit weight (*γ*_L_), pH, and EC, measured at the room temperature (22.5 to 25 °C), of the four chemical solutions. Landfill leachate was also used to investigate the potential performance of CMC-CaB as applied in GCLs. The leachate was collected from the Zion Landfill (Zion, IL, USA). The pH, *EC*, and BOD_5_ of the leachate were measured as 7.27, 1048 mS/cm, and 160 mg/L, respectively, and the total suspended solids are less than 40 mg/L.

### 2.4. Properties of CMC-SACaB

The properties of CMC-treated bentonites include liquid limit, plastic limit, specific gravity, swell index, organic content, and pH. The liquid limit and plastic limit were determined using the one-point liquid limit method and hand method, in accordance with ASTM D4318 [33]. The specific gravity was determined as per ASTM D854 [34]. The detailed procedure for bentonite hydration before specific gravity determination is in accordance with Yang et al. [11], and three trials with variation of coefficient lower than 0.46% were conducted for each CMC-treated bentonite to ensure the accuracy. The swell index of bentonites in DIW was measured as per ASTM D5890 [37]. Although the boiling point of CMC is not available, the CMC melting point and auto-ignition temperature are 274 °C (decomposes) and 370 °C, respectively. Thus, the organic content was determined using the loss-on-ignition method with a furnace temperature of 440 °C according to the Testing Method C in ASTM D2974 [30]. The pH of bentonites was measured according to ASTM D4972 [36].

### 2.5. Swell Index with Chemical Solutions

A series of swell index tests were conducted on CMC-SACaB as per ASTM D5890 [37]. The metal concentrations of Pb-Zn, Cr(VI), and Ca solutions used for swell index test were set at 50, 100, and 500 mmol/L. It is noted that the metal concentration lower than 10 mmol/L was not considered in this study. This is due to the fact that inorganic salt solution with metal concentration lower than 10 mmol/L would lead to bentonite turbidity, which results in an underestimation of the swell index [41].

### 2.6. Hydraulic Conductivity Via Modified Fluid Loss Tests

The MFL test procedure followed was in accordance with Chung and Daniel [27]. Bentonite content in the slurry (*BC*) and applied overall pressure (*P*_0_) of the MFL test were controlled, and both hydraulic conductivity (*k*_MFL_, m/s) and corresponding void ratio were presented. These were conducted for the purpose of hydraulic conductivity comparison. A predetermined mass of bentonite was mixed with 350 mL chemical solution to form a bentonite-chemical solution slurry for 5 min. using a paddle mixer in a 500-mL plastic bottle. The slurry was then mixed and hydrated using a shaker for 24 to 48 h. The *BC* value was set at 6% for all samples, as suggested by Chung and Daniel [27]. The *BC* value is defined using the following equation:(1)$$BC=\frac{m_{\mathrm{Ben}}}{\left(m_{\mathrm{Ben}}+m_{\mathrm{cs}}\right)}$$
where *m*_Ben_ is the dry mass of bentonite and *m*_CS_ is the mass of chemical solution. The pH of the slurry was measured prior to the MFL test via API 13B-1 using a Thermo Orion 720 A pH meter. The *P*_0_ value was set at 400 kPa during the MFL test. In addition, the tests were also performed under *P*_0_ of 50, 100, 200, and 400 kPa to investigate the relationship between the average void ratio of bentonite (*e*_ave_) and hydraulic conductivity. Six filtrate volumes with 10-min. interval were measured within one hour during the test to evaluate *k*_MFL_ using Equation (2):(2)$$k_{\mathrm{MFL}}=\frac{\beta \gamma_{L}V_{t}^{2}}{2P_{0}A^{2}t}\times 10^{-4}=\frac{\beta \gamma_{L}}{2A^{2}\varphi }\times 10^{-4}$$
where *γ*_L_ is the unit weight of chemical solution (kN/m^3^); *V*_t_ is the filtrate volume over time *t* (mL); *P*_0_ is the applied overall pressure, equaling to the applied air pressure in this study (kPa); *A* is the cross-sectional area of the bentonite filter cake (cm^2^); *φ* is the slope of the *p*_0_*V*_t_/*t*-*V*_t_ relation curve (kPa·s/cm^6^); and, *β* is calculated, as follows:(3)$$\beta =\frac{BC\times \rho_{L}\left(1+e_{\mathrm{ave}}\right)}{\left(1-BC\right)\times \rho_{s}-e_{\mathrm{ave}}\times BC\times \rho_{L}}$$
where *BC* is the bentonite content of the slurry (%); *ρ*_L_ and *ρ*_S_ are the densities of chemical solution and bentonite (g/cm^3^), respectively; and, *e*_ave_ is the average void ratio of the bentonite filter cake. To determine the *e*_ave_ value, it is assumed that the bentonite filter cake was saturated and the CMC content was not changed during water content measurement.

At the end of the test, a thin layer of jelly-like bentonite-chemical solution mixture on the top of the bentonite filter cake was carefully removed by a scraper. Subsequently, the bentonite filter cake was placed in a heavy-duty aluminum container for water content measurement. After that, the oven-dried bentonite filter cake was ground into powder for determination of specific gravity as per ASTM D854 [34]. The hydraulic conductivity of the bentonite is calculated using Equation (2). The *P*_0_ value in Equation (2) is taken as the value of the air pressure, as suggested by Liu et al. [42].

The chemical solutions used in the MFL tests include Pb-Zn, Cr(VI), Ca solutions, and landfill leachate. The total metal concentrations of Pb-Zn, Cr(VI), or Ca solution were controlled as 0 (i.e., DIW), 1, 5, 10, and 20 mmol/L. The MFL testing program consisted of two parts: (1) MFL tests were conducted on CMC-SACaB with various CMC contents using Pb-Zn solution to investigate the influence of CMC content on hydraulic conductivity, and determine the optimized CMC content (*CC*_B, opt_), and (2) MFL tests were conducted on CMC-SACaB with CMC content of *CC*_B, opt_ using Cr(VI), Ca solutions and landfill leachate to investigate the influence of contaminant type on hydraulic conductivity. Table 3 presents the entire MFL testing program.

### 2.7. Hydraulic Conductivity Based on Flexible-Wall Permeameter Tests

The constant head hydraulic conductivity tests on CMC-SACaB and SACaB were conducted using the flexible-wall permeameter test (FWP test) according to ASTM D5084 [43]. DIW was used as a permeating liquid in all tests to compare the resulting *k* of bentonites with that obtained from the MFL tests. The hydraulic gradient ranging from 100 to 200 was applied, as it is typically used for bentonites and GCLs [14]. The initial water content of the sample was adjusted to the same value determined from the bentonite cake after the MFL test. The initial height of the samples was controlled to 7.5 cm, and the diameter was either 3.6 or 7.3 cm. The samples were consolidated under an isotropic stress of 13.8 kPa (2 psi) for two days, and then the effective confining stress of 6.9 kPa (1 psi) was applied during the hydraulic conductivity testing stage. This was done to avoid a significant difference between the final void ratio and the corresponding initial void ratio, since isotropic stress in the consolidation stage is larger than that in the permeation stage. A custom-fabricated acrylic cylinder was used to avoid any potential sample disturbance and inclination [11].

## 3. Results and Discussion

### 3.1. Properties of CMC-SACaB

Figure 2 presents the variation of pH with CMC content in DIW at 60 °C. The results show that the pH slightly decreases with the increase in CMC content in water and it stabilizes when the CMC content is higher than 0.6%. Table 4 presents the properties of the CMC-SACaBs and then compares them with those of HYPER clays reported in previous studies [18,44], which include organic content, plastic limit, specific gravity, and pH. Figure 3 presents the variations in liquid limit and swell index with CMC content in bentonite, as both index properties generally reflect the changing trends of hydraulic conductivity of bentonite. 

The results indicate that the organic content, liquid limit, plastic limit, and swell index all tend to increase as the CMC content increases, which is attributed to the intercalation of CMC chains into clay mineral sheet that leads to a more open interlayer between the bentonite particles [23]. The mechanism of such intercalation is attributed to the sorption of anionic CMC chains on clay particle surface through fixation [45]. The liquid limit and swell index of CMC-SACaB with CMC content less than 10% are lower than those of the HYPER clays. However, CMC contents of 10% and 14% result in a comparable liquid limit and swell index than those of the HYPER clay which has a CMC content of 2% to 16% in Na-bentonite, as reported in previous studies [9,18,21,44]. The liquid limit and swell index of the CMC10 sample increase by 110% and 164% as compared with those of SACaB (i.e., CMC0), and the increments of these two properties of CMC14 sample are 143% and 264%, respectively. On the other hand, specific gravity and pH slightly decrease with an increase in the CMC content, as shown in Table 4, which is attributed to the nature of CMC, i.e., the relatively lower pH (see Figure 2) and specific gravity of CMC when compared with the SACaB. It is reported that the typical specific gravity value of CMC is 1.59 [45].

### 3.2. Swell Index Exposed to Chemical Solutions

Figure 4 shows the relationship between the total metal concentration and swell index for Ca, Pb-Zn, and Cr(VI) solutions. The results indicate that the swell index increases with increasing CMC content when the metal concentration is lower than 100 mmol/L. In this range of metal concentration, the swell index of CMC-SACaBs is higher than that of SACaB. Meanwhile, a negligible difference in the swell index between CMC-SACaB and SACaB is found for chemical solutions with metal concentration of 500 mmol/L, and the swell index values of both CMC-SACaBs and SACaB decrease to less than 7.5 mL/2 g. The CMC-SACaBs with various CMC contents exhibit a considerable decrease in the swell index with increasing metal concentration. In addition, the swell index with the Ca solution is similar to that obtained with Pb-Zn and Cr(VI) solutions for a given CMC content and metal concentration. The swell index is also found to considerably decrease with the metal concentration of Cr(VI) solution, which is attributed to that: (1) potassium ion (K^+^) from the source chemical in Cr(VI) solution, whose molar concentration is twice of Pb and Zn in Pb-Zn solution or Ca in Ca solution, causes the compression of the diffuse double layer of bentonite particles, i.e., a decrease in the swell index; and, (2) bentonite is allowed to swell freely without confining pressure, which can, to some extent, offset the influence of the chemical solution on the hydraulic conductivity of clayey soils.

Figure 5 presents the variation in swell index ratio (*SIR*) with CMC content in bentonite and metal concentration to better understand the swell potential of the CMC-treated bentonites. The *SIR* is defined as the ratio of the swell index of sample in chemical solution to that in DIW. The results of HYPER clay (*CC*_B_ = 2%) with Ca solution that were reported by Di Emidio et al. [18] are also presented for comparison. It is evident that the *SIR* decreases with an increasing metal concentration. CMC-SACaB possesses a comparable *SIR* value when exposed to Ca solution when compared to HYPER clay. The *SIR* values of CMC-SACaBs with CMC contents that were tested in this study range from 0.3 to 0.4 and 0.08 to 0.19 when Ca concentration is 100 and 500 mmol/L, respectively, whilst they are 0.24 and 0.17 for HYPER clay under the same conditions. However, CMC-SACaB and HYPER clay both do not show higher *SIR* values than the corresponding untreated bentonites. A slight difference in *SIR* is found between CMC-SACaBs and SACaB with metal concentration of 50 mmol/L (*SIR* = 0.5 to 0.6); while the decreasing trend of *SIR* with CMC content at metal concentrations of 100 and 500 mmol/L is more noticeable. Di Emidio et al. [18] reported similar results, where the *SIR* value of HYPER clay is 10% to 40% lower than that of untreated Na-bentonite, and this difference in *SIR* value is more noticeable at relatively high metal concentration.

### 3.3. Hydraulic Conductivity

Figure 6 illustrates the typical relationship between *P*_0_·*t*/*V*_t_ and *V*_t_ using a CMC10 sample that is exposed to DIW and ionic solutions with metal concentration of 10 mmol/L. The results show that all *P*_0_·*t*/*V*_t_–*V*_t_ curves are linear. Thus, the parameter *φ* in Equation (2) can be determined based on the slope of the *P*_0_·*t*/*V*_t_ and *V*_t_ relationship using the Least-Square-Root method. The coefficient of determination (*R*^2^) for all of the samples tested ranges from 0.990 to 0.999.

Figure 7 presents the relationship between metal concentration and the hydraulic conductivity obtained from the MFL tests (*k*_MFL_). The CMC treatment results in a considerable decrease in *k*_MFL_ as compared to the SACaB for a given type of metal and metal concentration. This result is more noticeable with increasing metal concentration. For Pb-Zn solution with metal concentration lower than 10 mmol/L, the *k*_MFL_ of CMC-SACaBs with all of the *CC*_B_ values used (*CC*_B_ = 2% to 14%) is consistent with the corresponding *k*_MFL_ evaluated using DIW, and it tends to increase when the metal concentration increases to 20 mmol/L. In contrast, the *k*_MFL_ of the SACaB starts to increase at a metal concentration of 1 mmol/L. For a metal concentration of 20 mmol/L, the *k*_MFL_ of CMC6 is slightly higher than that of CMC2, which is due to a higher void ratio of the former, as shown in Figure 7. It is important to note that CMC10 and CMC14 bentonites possess consistent *k*_MFL_ value when exposed to Pb-Zn solution. Consequently, the optimized CMC content in bentonite is concluded to be 10% for the SACaB that was used in this study. On the other hand, the results indicate that the Ca and Cr(VI) solutions with metal concentrations of 20 mmol/L have noticeably less influence on the *k*_MFL_ of CMC10 bentonite than the Pb-Zn solutions. For the Cr(VI) solution, it is reported that the original dispersed fabric of bentonite particles is unlikely to be affected by the presence of Cr(VI) as an anionic complex when overburden confining pressure is applied on the CMC10 sample during the MFL tests, regardless of the possible cation exchange reactions between K ions in Cr(VI) solution and the exchangeable cations in the DDL of CMC-SACaB particles [42].

Figure 8 presents the relationship between the average void ratio (*e*_ave_) and *k*_MFL_ for CMC0 and CMC10 samples, in which metal concentration is 10 mmol/L, to better understand the impact of metal contamination on the *k*_MFL_ of CMC-SACaB. The increase in *k*_MFL_ at a given *e*_ave_ due to exposure to chemical solution for CMC0 is evident. For example, the *k*_MFL_ of the CMC0 sample corresponding to *e*_ave_ of 7.1 increases 413 times when exposed to a Pb-Zn solution. In contrast, an insignificant increase in *k*_MFL_ is found for CMC10 when exposed to various chemical solutions. The ratio of the hydraulic conductivity permeated with chemical solution to that permeated with DIW (*HCR*) corresponding to *e*_ave_ of 13.8 to 14.4 is 1.5, 2.2, and 1.3 for the Pb-Zn, Cr(VI), and CaCl_2_ solutions, respectively. The *HCR* is defined as follows:(4)$$HCR=\frac{k_{c}}{k_{w}}$$
where *k*_c_ and *k*_w_ are hydraulic conductivity permeated with chemical solution and DIW, respectively.

It is understood that the hydraulic conductivity for untreated bentonites tends to increase when permeated with heavy metals-laden water. This is because of the reduced diffused double layer (DDL) thickness of bentonite, which resulted from the ion exchange reactions between metal (e.g., Pb, Zn, and Ca) ions in chemical solutions and exchangeable cations initially adsorbed on the bentonite particles (e.g., sodium and calcium ions). The degree of contraction of DDL increases with the increase in metal concentration, based on the Gouy–Chapman theory [5]. The benefit of the CMC treatment is mainly attributed to the chemical interaction between carboxylate (COO-) groups from CMC chains and cation (i.e., K^+^, Ca^2+^, Pb^2+^, Zn^2+^, and Cr^3+^) from chemical solutions in the form of either unidentate, bidentate, or bridging [46]. In addition, Na^+^ from hydrated CMC chains can be adsorbed on SACaB through ion exchange with exchangeable Ca^2+^ (14.5 cmol*_c_*/kg, as presented in Table 1) initially adsorbed on the SACaB particles, promoting further sodium treatment [10,11]. Consequently, the DDL of the bentonite particles is protected from compression when heavy metals in the source chemical solution are ready to react with CMC chains. However, the exhaustion of the CMC chains implies an increase in *k*_MFL_ and a decrease in the swell potential, e.g., different degrees of increase in *k*_MFL_ and decrease in the swell index of CMC-treated bentonites when exposed to an ionic solution with a metal concentration greater than 10 mmol/L (see Figure 3 and Figure 7).

Figure 9 presents the impact of landfill leachate on the hydraulic conductivity of CMC-SACaBs and SACaB. In addition, Figure 9 presents the average void ratio of bentonite and the pH of the bentonite-leachate slurry. The results show that the *k*_MFL_ tends to decrease with an increase in *CC*_B_. The *k*_MFL_ value of CMC2, CMC6, CMC10, and CMC14 decreases 12, 38, 48, and 22 times when compared with that of SACaB. The relatively higher *k* of CMC14 when compared with that of CMC10 is due to its corresponding higher *e*_ave_ (*e*_ave_ of 3.31 for CMC10 and 4.90 for CMC14). Thus, it can be concluded that CMC treatment significantly contributes to the decrease in *k*_MFL_ of SACaB when exposed to landfill leachate. In addition, the slurry pH slightly increases with increasing *CC*_B_, and it stabilizes at approximately 7.28, which is almost the same with the leachate pH value (7.26).

Liu et al. [42] indicated that there existed a unique relationship between filtrate volume after 30 min. (*FV*) and hydraulic conductivity obtained from the MFL tests for both clean and heavy metal (lead, cadmium, and hexavalent chromium) contaminated bentonites under a given applied overall pressure, which can be expressed by Equation (5):(5)$$\log\left(k_{MFL}\right)=A\log\left(FV\right)-B$$
where the slope *A* and intercept *B* varies from 1.32 to 1.43 and 11.1 to 12.5, respectively, when the applied overall pressure increases from 50 kPa to 690 kPa, indicating that the *FV*–*k*_MFL_ relationship is crucially affected by *P*_0_. On the other hand, the *HCR* is considered to be an effective indicator that reflects the effect of metal solution on *k*_MFL_ of bentonite [5,39]. An attempt is made to correlate the *HCR* with filtrate volume ratio (*FVR*) under various applied overall pressure (50 to 690 kPa) in this study and previous study [42]. The *F**VR* is defined, as follows:(6)$$FVR=\frac{FV_{c}}{FV_{w}}$$
where *FV*_c_ and *FV*_w_ are filtrate volume after 30 min. of sample exposed to chemical solution (i.e., contaminated sample) and DIW (i.e., clean sample), respectively.

A unique relationship exists between *HCR* and *FVR* on a double logarithmic scale as shown in Figure 10. In addition, the overall trend of the *HCR-FVR* relationship of the CMC-treated bentonites is in line with that of untreated Ca- and Na-bentonites. The empirical relationship determined using a Least-Square-Root method could be expressed by Equation (7) with the coefficient of determination (*R*^2^) of 0.883. A further statistical analysis indicates that the residual sum of squares is 3.733, and 95% confidence and prediction bands are also presented using shadow zones, as shown in Figure 10.
(7)log(HCR)=1.63log(FVR)−0.0183

Table 5 presents the comparison of hydraulic conductivity measured using the MFL tests with that measured using DIW as permeating liquid in the FWP tests. Table 5 lists the corresponding average void ratio of the bentonite in the MFL tests and the final void ratio of the bentonite sample in the FWP tests. The results show that the *k* values estimated from the MFL tests are consistent with those that determined using the FWP tests under a similar range of void ratios. Similar results were reported by Chung and Daniel [27] and Liu et al. [29], in which conventional Na-bentonites were used. Thus, it is concluded that the MFL test can be used as a quick method for measuring the *k* of bentonites based on the results from this study and previous studies.

### 3.4. Study Limitations

The experience from this study indicates that the MFL test is not appropriate for measuring the *k*_MFL_ of bentonite under relatively high metal concentration conditions. The prepared bentonite–chemical solution slurry for the MFL test should maintain a state of suspension, and potential solid-liquid separation due to the aggregation of bentonite particles should not be allowed. This is because the bentonite filter cakes formed by aggregation sedimentation may result in either a gap between the bentonite filter cake and API cell or cracks in the bentonite filter cake, which may lead to an overestimated hydraulic conductivity. For example, the metal concentration of Pb-Zn solution should not exceed 20 mmol/L for all of the CMC-SACaB and SACaB samples tested in this study. Figure 11 illustrates the typical defective bentonite filter cakes formed in bentonite/Pb-Zn solution with metal concentrations of 50 and 100 mmol/L. It is found that the bentonite thickness of both bentonite filter cakes shown in Figure 11 can be as thick as 40 mm due to bentonite particle aggregation when exposed to Pb-Zn solution with metal concentrations of both 50 and 100 mmol/L. The formation of aggregation structure of bentonite particles is also addressed in the authors’ previous study [41]. In contrast, the bentonite thickness of bentonite filter cakes formed in Pb-Zn solution with metal concentration lower than 20 mmol/L is found to be no more than 5 mm in this study. Moreover, three cracks can be clearly observed at the bottom of CMC2 bentonite filter cake in Pb-Zn solution with total metal concentration of 50 mmol/L. Further study is warranted to systematically compare the results obtained from the MFL test with those obtained from long-term FWP test permeated various chemical solutions.

In this study, a series of tests were conducted to evaluate the benefit of CMC treatment on the index properties, hydraulic conductivity, and contaminant-compatibility of CMC-SACaB exposed to heavy metals-laden water and actual landfill leachate. It is noted that analyses of contaminant transport and long-term performance are important in the design of engineered barriers. Therefore, further studies are warranted to investigate contaminant transport parameters (e.g., effective diffusion coefficient and retardation factor) and long-term performance in terms of long-term hydraulic conductivity, potential elution of CMC, mechanical properties, and wet-dry cycle durability [47,48,49,50,51,52,53,54,55,56,57,58,59]. In addition, X-ray diffraction, Zeta potential, and Fourier transform infrared spectroscopy analyses are recommended in the further studies. These micro-analyses are useful in understanding changes in clay minerals and microstructures of CMC-SACaB and metal-CMC interactions exposed to the chemical solutions. 

## 4. Conclusions

This study investigated the properties and chemical compatibility of CMC-SACaB with ionic solutions. The results were also compared with those that were obtained from CMC-treated Na-bentonite (i.e., HYPER clay) reported in published studies. The following conclusions can be drawn:

The liquid limit, plastic limit, and swell index of the CMC-SACaB increased with an increase in the CMC content in bentonite, which was attributed to the intercalation of the CMC chains into the clay sheet. However, the specific gravity and pH of CMC-SACaB slightly decreased with an increase in the CMC content in bentonite due to the nature of the CMC. The CMC-SACaB possessed a comparable swell index with that of HYPER clay when the CMC content in Ca-bentonite was 10%. 

The swell index of CMC-SACaBs with Pb-Zn, Cr(VI), and Ca solutions was considerably higher than that of SACaB. However, CMC-treatment was not able to promote the swell potential under a high metal concentration (e.g., 500 mmol/L). CMC-SACaB under Ca solution with metal concentrations of 50 and 100 mmol/L showed a similar swell index ratio when compared with that of HYPER clay, which indicated comparable chemical compatibility in terms of swell potential. However, the swell index ratio of the CMC-SACaB was lower than that of HYPER clay when the Ca concentration was 500 mmol/L.

The results from MFL tests indicated that CMC-SACaBs possessed a lower hydraulic conductivity with Pb-Zn, Cr(VI), Ca solutions, and landfill leachate when compared to the SACaB. An insignificant change in *k*_MFL_ with metal concentration for CMC-SACaB under Pb-Zn solution was found until the metal concentration was higher than 10 mmol/L. Cr(VI) and Ca solutions with metal concentrations lower than 20 mmol/L did not affect the *k*_MFL_ of CMC-SACaBs whose CMC content was 10%. The optimum CMC content in bentonite was found to be 10% as the resulting *k*_MFL_ of CMC-SACaB with 14% CMC under Pb-Zn solution was the same with that of CMC-SACaB containing 10% CMC. The beneficial effect of CMC-treatment was attributed to the chemical interaction between the carboxylate (COO-) groups from CMC chain and cations from chemical solutions and bentonite sodium treatment by Na ion from CMC. 

There was a unique relationship between the filtrate volume ratio and the hydraulic conductivity ratio for various types of bentonites and chemical liquids. Careful observation was needed after bentonite-chemical slurry preparation, and the slurries that showed noticeable bentonite aggregation sedimentation could not be used for MFL test. This study suggested that an inorganic solution with a total metal concentration higher than 20 mmol/L was unsuitable for the MFL test.

## Figures and Tables

**Figure 1 ijerph-17-01863-f001:**
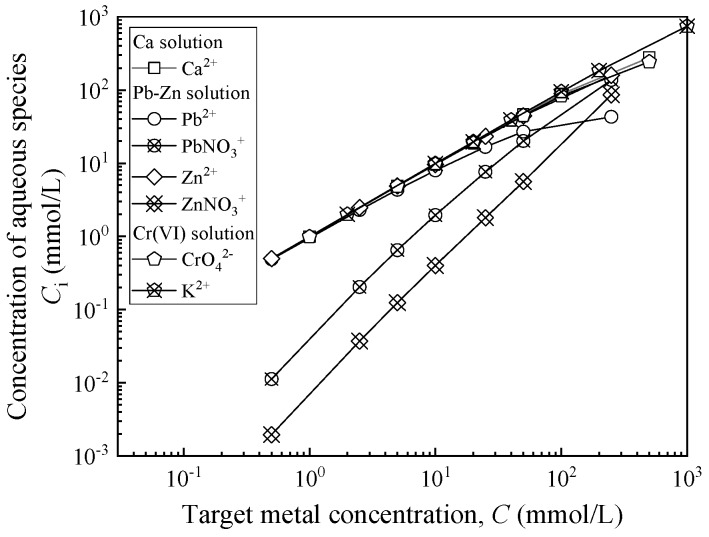
Comparison between concentration of aqueous species and total metal concentration.

**Figure 2 ijerph-17-01863-f002:**
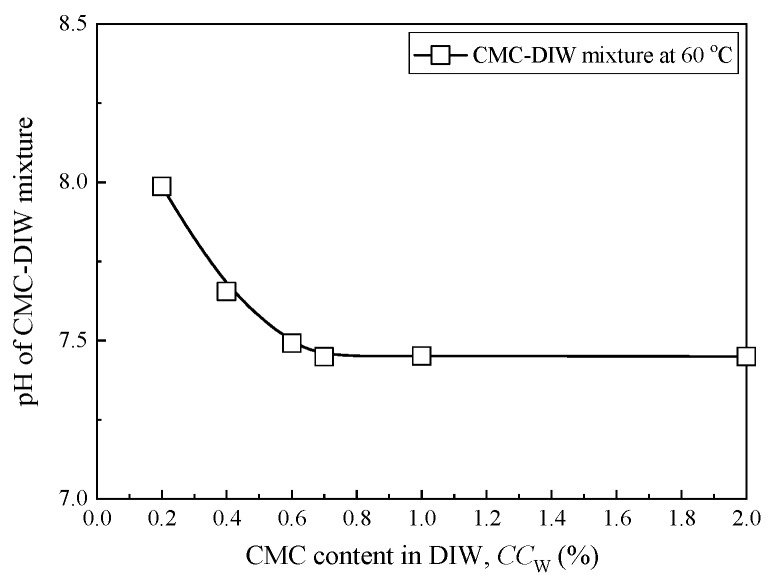
Relationship between carboxymethyl cellulose (CMC) content in deionized water (DIW) and pH.

**Figure 3 ijerph-17-01863-f003:**
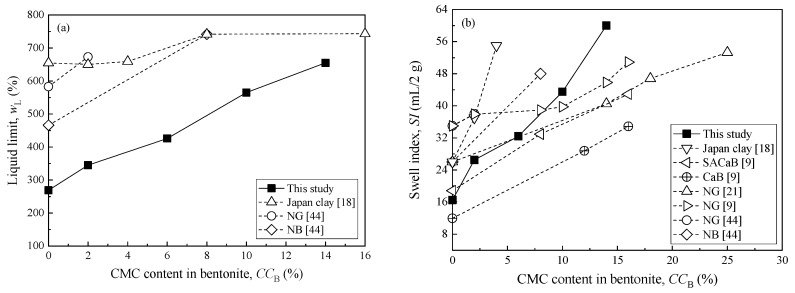
Relationship between CMC content in bentonite (**a**) and swell index (**b**) using DIW (Note: CaB, Japan clay, SACaB, NG, NB are base soils used by previous studies).

**Figure 4 ijerph-17-01863-f004:**
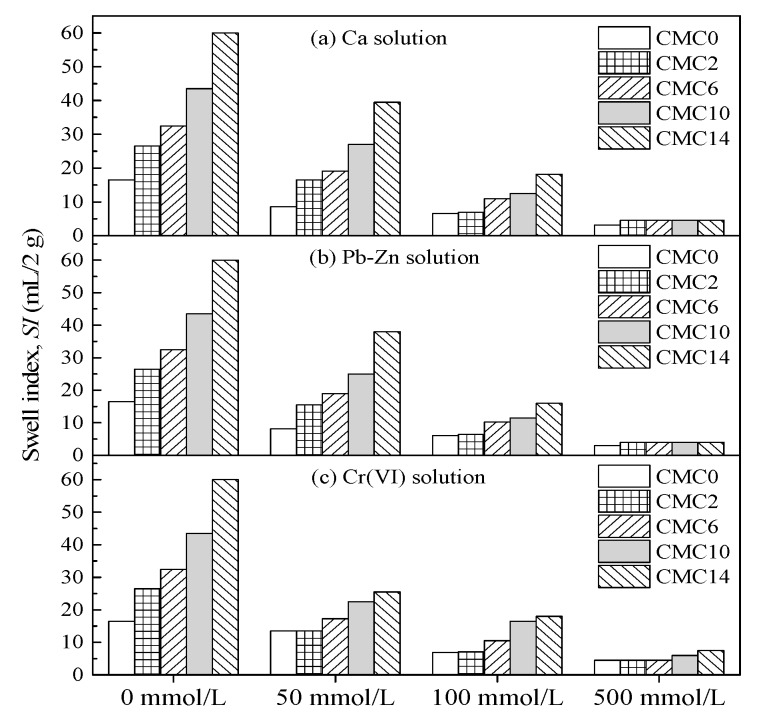
Change of swell index with metal concentration in chemical solutions: (**a**) Ca solution; (**b**) Pb-Zn solution; and, (**c**) Cr(VI) solution.

**Figure 5 ijerph-17-01863-f005:**
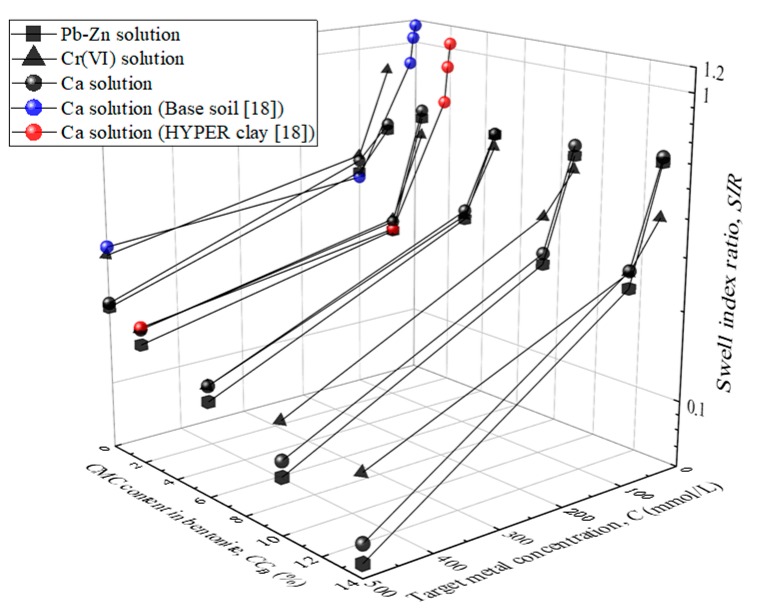
Variation in swell index ratio (*SIR*) with CMC content in bentonite (*CC*_B_) and metal concentration (*C*).

**Figure 6 ijerph-17-01863-f006:**
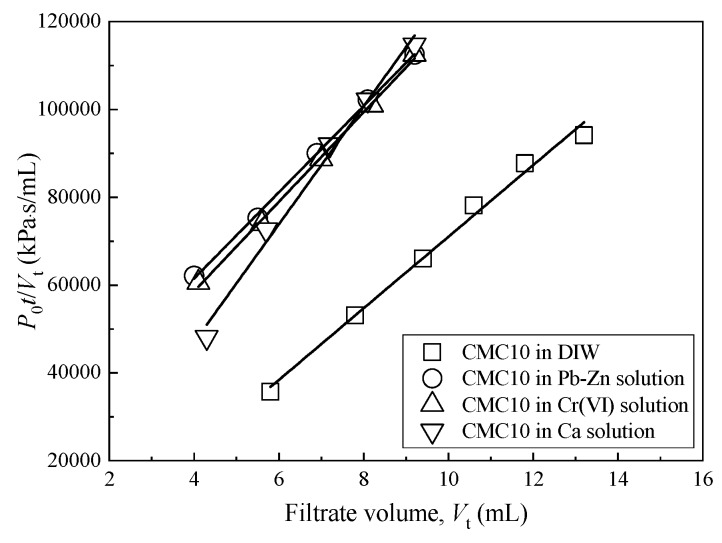
Relationship between *P*_0_·*t*/*V*_t_ and *V*_t_ for CMC10 samples that were exposed to DIW and ionic solutions with metal concentration of 10 mmol/L.

**Figure 7 ijerph-17-01863-f007:**
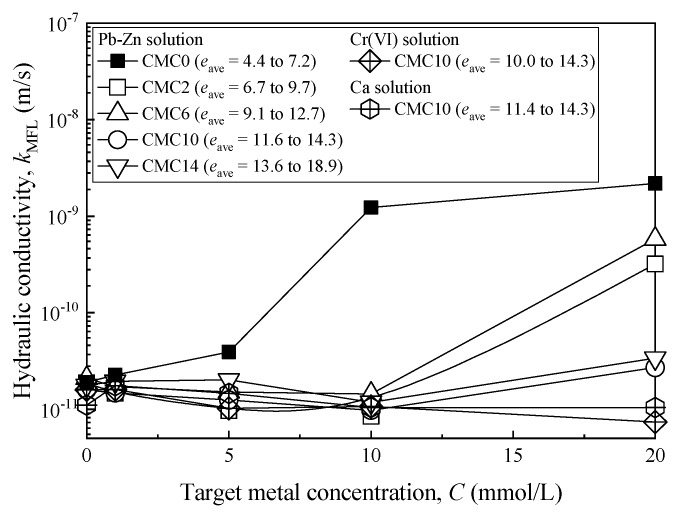
Relationship between metal concentration (*C*) and hydraulic conductivity (*k*_MFL_) for samples exposed to Pb-Zn, Cr(VI) and Ca solution.

**Figure 8 ijerph-17-01863-f008:**
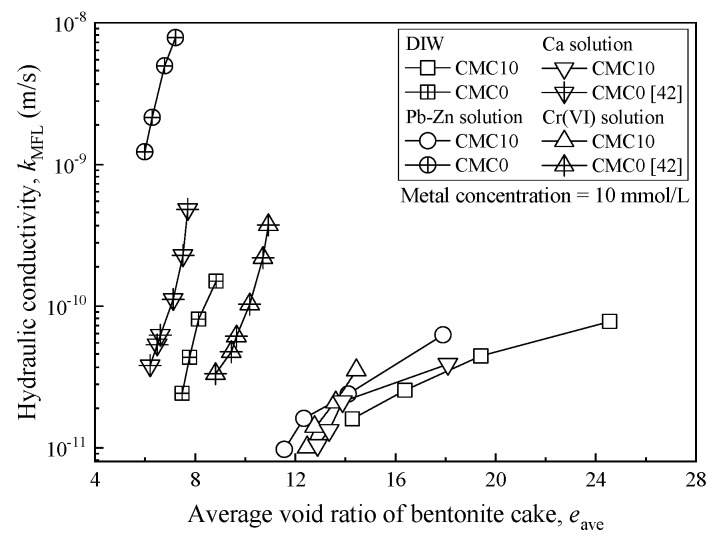
Relationship between average void ratio and hydraulic conductivity for sample exposed to Pb-Zn, Cr(VI) and Ca solution.

**Figure 9 ijerph-17-01863-f009:**
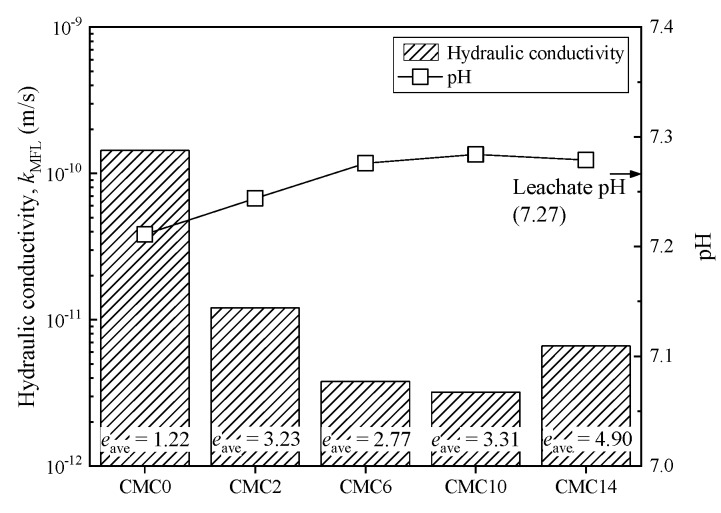
Variation in hydraulic conductivity and pH with CMC content in bentonite for samples exposed to landfill leachate.

**Figure 10 ijerph-17-01863-f010:**
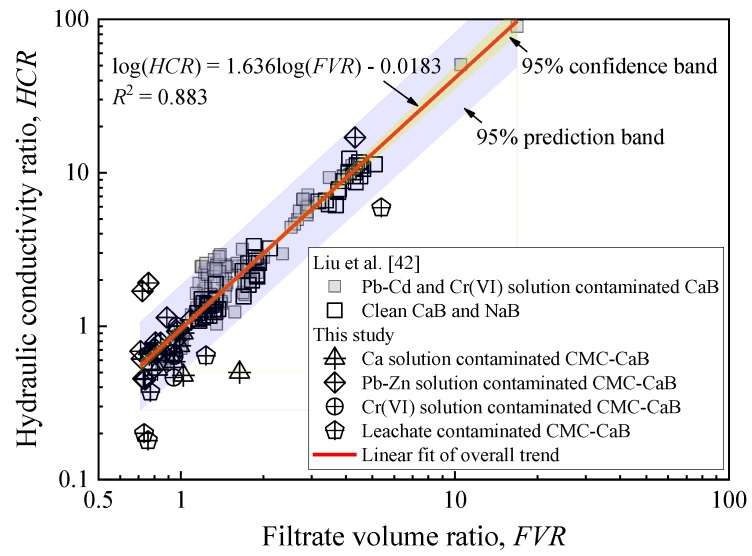
Relationship between filtrate volume ratio and hydraulic conductivity ratio obtained from MFL test.

**Figure 11 ijerph-17-01863-f011:**
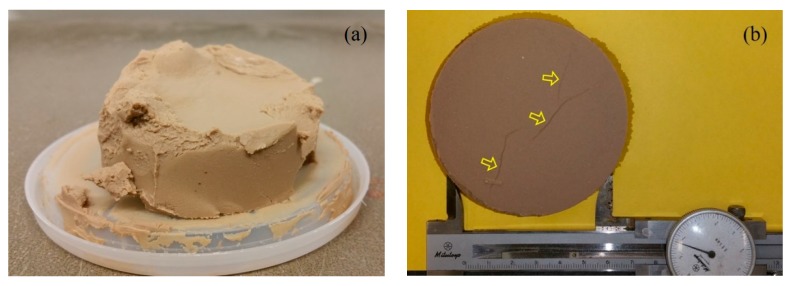
Observed bentonite filter cakes formed under relatively high metal concentration in Pb-Zn solutions: (**a**) CMC10 bentonite filter cake under Pb-Zn solution with metal concentration of 100 mmol/L and (**b**) CMC2 bentonite filter cake under Pb-Zn solution with metal concentration of 50 mmol/L.

**Table 1 ijerph-17-01863-t001:** Properties of sodium activated calcium bentonite (SACaB).

Property	Testing Method	Value
Fines Content, *FC* (%)	[31]	100
Clay Content, *CC* (%)	[31]	49
Organic Content, *OC* (%)	[32]	1.18
Liquid Limit, *w*_L_ (%)	[33]	269
Plastic Limit, *w*_P_ (%)	[33]	34.0
Specific Gravity, *G*_s_	[34]	2.66
Classification	[35]	CH
pH	[36]	10.33
Swell index, *SI* (mL/2 g)	[37]	16.5
Exchangeable metals (cmol*_c_*/kg):	[38]	
Ca^2+^		14.5
Mg^2+^		0.6
Na^+^		32.7
K^+^		0.3

**Table 2 ijerph-17-01863-t002:** Relevant properties of chemical solutions measured at room temperature (22.5 to 25 °C).

Chemical Solution	Total Metal Concentration, *C* (mmol/L)	Ionic Strength, *I* (mmol/L)	Absolute Viscosity, *μ* ^a^ (mPa·s)	Unit Weight, *γ*_L_ (kN/m^3^)	pH	Electrical Conductivity, *EC* (mS/cm)
DIW	N.D.	-	1.00	9.79	6.90	0.004
Ca solution	1	2.99	1.02	9.79	6.80	0.27
Ca solution	5	14.80	1.03	9.81	6.79	1.21
Ca solution	10	29.50	1.03	9.85	6.78	2.36
Ca solution	20	58.40	1.04	9.90	6.76	4.24
Ca solution	50	142.50	1.05	9.94	6.61	10.52
Ca solution	100	275.80	1.07	9.99	6.46	16.45
Ca solution	500	1444.83	1.15	10.46	5.90	72.8
Pb-Zn solution	1	2.97	1.26	9.79	5.49	0.31
Pb-Zn solution	5	14.50	1.28	9.81	5.39	1.22
Pb-Zn solution	10	28.40	1.31	9.85	5.31	2.20
Pb-Zn solution	20	55.00	1.38	9.90	5.21	3.86
Pb-Zn solution	50	129.10	1.43	9.93	5.09	8.28
Pb-Zn solution	100	239.60	1.54	10.01	4.94	15.66
Pb-Zn solution	500	829.00	1.69	10.87	4.33	51.15
Cr(VI) solution	1	2.98	1.08	9.79	7.69	0.29
Cr(VI) solution	5	14.80	1.09	9.85	8.26	1.38
Cr(VI) solution	10	29.20	1.10	9.88	8.40	2.58
Cr(VI) solution	20	57.60	1.10	9.92	8.69	4.76
Cr(VI) solution	50	139.30	1.14	9.99	8.87	9.71
Cr(VI) solution	100	266.60	1.18	10.07	9.20	16.32
Cr(VI) solution	500	984.70	1.35	10.59	9.33	78.4

N.D.: not determined. ^a^ Absolute viscosity is measured using a NDJ-5S model rotational viscometer with spindle of No.0.

**Table 3 ijerph-17-01863-t003:** Modified fluid loss (MFL) testing program.

CMC Content in Bentonite, *CC*_B_ (%)	Chemical Solution	Total Metal Concentration, *C* (mmol/L)	Applied Overall Pressure, *P*_0_ (kPa)
0, 2	Pb-Zn	0, 1, 5, 10, 20	400
6, 10, 14	Pb-Zn	0, 1, 5, 10, 20	400
Optimized *CC*_B_	Cr(VI), Ca	0, 1, 5, 10, 20	400
0, Optimized *CC*_B_	landfill leachate	-	400
0	Pb-Zn	0, 10	50, 100, 200, 400
Optimized *CC*_B_	Pb-Zn, Cr(VI), Ca	0, 10	50, 100, 200, 400

**Table 4 ijerph-17-01863-t004:** Properties of CMC-SACaB in this study and CMC-treated Na-bentonite reported in published studies.

Sample ID	CMC Content in Bentonite, *CC*_B_ (%)	Organic Content, *OC* (%)	Plastic Limit, *w*_P_ (%)	Specific Gravity, *G*_s_	pH	References
CMC2	2	2.88	38	2.63	9.55	This study
CMC6	6	5.77	52	2.61	9.43	This study
CMC10	10	10.39	55	2.61	9.26	This study
CMC14	14	13.99	56	2.59	9.21	This study
HC2	2	N.A.	73	N.A.	N.A.	[44]
HC8	8	N.A.	61	N.A.	N.A.	[44]
HC2%	2	N.A.	56.22	2.53	N.A.	[18]
HC4%	4	N.A.	58.62	2.47	N.A.	[18]
HC8%	8	N.A.	61.03	2.25	N.A.	[18]
HC16%	16	N.A.	70.55	2.40	N.A.	[18]

N.A.: not available; HC: hyper clay.

**Table 5 ijerph-17-01863-t005:** Comparison of hydraulic conductivity determined using MFL test with that measured using flexible wall permeameter (FWP) test.

Sample ID	MFL Test			FWP Test			
	*e* _ave_ ^1^	*ρ* _d_ ^3^	*k* _MFL_ ^4^	*e* _f_ ^2^	*ρ* _d_ ^3^	*i* ^5^	*k* _FWP_ ^6^
CMC0	7.4	0.32	2.5 × 10^−11^	6.6	0.35	94	6.1 × 10^−11^
CMC2	9.7	0.25	1.9 × 10^−11^	9.1	0.26	135	1.6 × 10^−11^
CMC6	16.7	0.15	2.1 × 10^−11^	15.3	0.16	172	1.0 × 10^−11^
CMC10	14.3	0.17	1.6 × 10^−11^	13.1	0.19	119	9.6 × 10^−12^
CMC14	18.9	0.13	1.8 × 10^−11^	17.8	0.14	137	8.6 × 10^−12^

^1^*e*_ave_ is average void ratio of bentonite sample from MFL test. ^2^
*e*_f_ is final void ratio of bentonite sample from FWP test. ^3^
*ρ*_d_ is dry density of bentonite sample in g/cm^3^, calculated using *G*_s_ shown in Table 1 and Table 4, void ratio shown in Table 5, and density of water (1.0 g/cm^3^). ^4^
*k*_MFL_ is hydraulic conductivity determined by MFL test in m/s. **^5^**
*i* is hydraulic gradient. ^6^
*k*_FWP_ is hydraulic conductivity determined by FWP test in m/s.

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
