# Peer review of "Index Properties, Hydraulic Conductivity and Contaminant-Compatibility of CMC-Treated Sodium Activated Calcium Bentonite"

_ijerph, 2020, doi:10.3390/ijerph17061863_

Round 1

Reviewer 1 Report

Line 45/46 – not a lower proportion – they are just different (ionts). They have lower “strength” but both types cause swelling

Line 96 – what does it mean improve hydraulic conductivity – increase or decrease?

table 5 - what was dry density of samples (I was not able to determine it from provided data)? For most bentonites hydraulic conductivity is exponential function of density. Provided values seems to be quite high comparing to other values in literature (measured by other tests). But without density they can't be compared.

General note: The paper provides overview of material performance but does not look what happened to the material. E.g. was there any change to (micro)structure? Chemical composition? Mineralogical changes? Was CMC consumed? What has leached out? What processes happened inside? These things could indicate on (long term) performance of sealing layers... And yes I know that this was out of scope of paper but that would give insight of what is happenning.

Author Response

The authors would like to thank the Editor and the Reviewers for their constructive comments and suggestions. The comments are all valuable and helpful for improving our revised manuscript. All of the review comments have been carefully considered and incorporated into the revised manuscript. Itemized responses to the reviewers’ comments and suggestions are provided below.

Please note that the reviewers’ comments are shown in normal font and our responses are shown in italic fonts in red below. In the revised manuscript, the revised sentences/sections are marked in red. In addition to the Editor’s and the Reviewers’ comments and suggestions, we also made some other revisions to improve the manuscript. These revisions do not influence the content or framework of the manuscript; hence here we do not list the changes but also mark in red in the revised manuscript.

Comment 1: Line 45/46 – not a lower proportion – they are just different (ionts). They have lower “strength” but both types cause swelling.

Authors’ reply: Thanks for the comment. We agree with the reviewer. CaB and SACaB show relatively lower swell potential when compared with NaB due to lower amount of exchangeable sodium ions. Please see Page 2, Line 46 to 47.

Comment 2: Line 96 – what does it mean improve hydraulic conductivity – increase or decrease?

Authors’ reply: Thanks for the reviewer’s careful check. What we meant is “reduce the hydraulic conductivity”. This has been modified in the revised manuscript, and please see Page 3, Line 97.

Comment 3: table 5 - what was dry density of samples (I was not able to determine it from provided data)? For most bentonites hydraulic conductivity is exponential function of density. Provided values seems to be quite high comparing to other values in literature (measured by other tests). But without density they can't be compared.

Authors’ reply: Please note that void ratio is extensively used to correlate hydraulic conductivity of clayey soils including bentonite, which is addressed in classic textbooks of soil mechanics and number of previously published papers. We provide values of dry density of samples in Table 5 of the revised manuscript. The dry density is calculated using specific gravity shown in Table 4, void ratio shown in Table 5, and density of water (1.0 g/cm3). Please see Page 14, Table 5.

General note: The paper provides overview of material performance but does not look what happened to the material. E.g. was there any change to (micro)structure? Chemical composition? Mineralogical changes? Was CMC consumed? What has leached out? What processes happened inside? These things could indicate on (long term) performance of sealing layers. And yes I know that this was out of scope of paper but that would give insight of what is happening.

Authors’ reply: We thank the reviewer for this constructive comment. We agree with the reviewer that changes in minerals, microstructure and CMC dosage change and elution (leaching) are beyond the scope of this study. We have conducted x-ray diffraction (XRD), zeta potential, and Fourier transform infrared (FTIR) spectroscopy analyses to investigate micro-mechanisms controlling the interactions between CMC and bentonite. The results are reported in the first author’s PhD thesis, and a manuscript addressing these micro-properties will be submitted to an international journal. In addition, long-term hydraulic conductivity on CMC-modified bentonites are warranted to investigate the potential elution of CMC. The authors have addressed these further studies in the section 3.4 of the revised manuscript. Please see Page 15, Line 451 to 461.

Reviewer 2 Report

Reviewed paper deals with quite interesting and present problem. After introduction part, in which authors good present the problem and present state of knowledge, the methodology is presented in clear way. Also results are presented in easy to follow way and compared with results obtained by other authors. Conclusions are accurate and backed by results. 

Author Response

The authors would like to thank the Editor and the Reviewers for their constructive comments and suggestions. The comments are all valuable and helpful for improving our revised manuscript. All of the review comments have been carefully considered and incorporated into the revised manuscript. Itemized responses to the reviewers’ comments and suggestions are provided below.

Please note that the reviewers’ comments are shown in normal font and our responses are shown in italic fonts in red below. In the revised manuscript, the revised sentences/sections are marked in red. In addition to the Editor’s and the Reviewers’ comments and suggestions, we also made some other revisions to improve the manuscript. These revisions do not influence the content or framework of the manuscript; hence here we do not list the changes but also mark in red in the revised manuscript.

Comments and Suggestions for Authors: Reviewed paper deals with quite interesting and present problem. After introduction part, in which authors good present the problem and present state of knowledge, the methodology is presented in clear way. Also results are presented in easy to follow way and compared with results obtained by other authors. Conclusions are accurate and backed by results.

Authors’ reply: We thank the reviewer for recognizing the value of our research.

Reviewer 3 Report

This manuscript introduced the index properties, hydraulic conductivity and contaminant-compatibility of CMC-treated sodium activated calcium bentonite. The topic of this study is interesting. But some parts still need to be improved. Here are some specific comments.
(1) The literature review needs to be more critical.
(2) Why did the author focus on such index in this study?
(3) The results can be impacted by many testing conditions. Please explain.
(4) More details about statistical analysis in result analysis is required.
(5) Error bars are required in some figures.
(6) Please review and add the following two refences about the restriction of contaminant movement.
Bai, X., Song, K., Liu, J., Mohamed, A. K., Mou, C., & Liu, D. (2019). Health Risk Assessment of Groundwater Contaminated by Oil Pollutants Based on Numerical Modeling. International journal of environmental research and public health, 16(18), 3245.
Zhao, S., Huang, G., An, C., Wei, J., & Yao, Y. (2015). Enhancement of soil retention for phenanthrene in binary cationic gemini and nonionic surfactant mixtures: Characterizing two-step adsorption and partition processes through experimental and modeling approaches. Journal of hazardous materials, 286, 144-151.
(7) Please compare the results in this study with those in previous studies.
(8) The language also needs to be improved.

Author Response

The authors would like to thank the Editor and the Reviewers for their constructive comments and suggestions. The comments are all valuable and helpful for improving our revised manuscript. All of the review comments have been carefully considered and incorporated into the revised manuscript. Itemized responses to the reviewers’ comments and suggestions are provided below.

Please note that the reviewers’ comments are shown in normal font and our responses are shown in italic fonts in red below. In the revised manuscript, the revised sentences/sections are marked in red. In addition to the Editor’s and the Reviewers’ comments and suggestions, we also made some other revisions to improve the manuscript. These revisions do not influence the content or framework of the manuscript; hence here we do not list the changes but also mark in red in the revised manuscript.

Comments and Suggestions for Authors: This manuscript introduced the index properties, hydraulic conductivity and contaminant-compatibility of CMC-treated sodium activated calcium bentonite. The topic of this study is interesting. But some parts still need to be improved.

Authors’ reply: We thank the reviewer for the constructive comments. All of the reviewer’s comments are incorporated in the revised manuscript.

Comment 1: The literature review needs to be more critical.

Authors’ reply: Thanks for this comment. A review on the importance of index properties on hydraulic conductivity and contaminant-compatibility of clayey soils are presented in Section 1. Please see Page 3, Line 103 to 111.

Comment 2: Why did the author focus on such index in this study?

Authors’ reply: Index properties, including liquid limit, plastic limit, specific gravity and swell index, are fundamental properties of bentonites (Yang et al. 2018). Moreover, liquid limit and swell index can be considered as basis for initial assessment of hydraulic conductivity and contaminant-compatibility of clayey soils being potentially used as engineered barrier materials (Mitchell and Soga 2005). For instance, a bentonite with relatively higher values of liquid limit or swell index generally possesses lower hydraulic conductivity at a given void ratio. Hence, liquid limit, swell index, and void ratio are extensively used to establish models for predicting hydraulic conductivity of fine-grained soils (e.g., Chapuis 2012). In addition, liquid limit ratio, defined as the ratio of liquid limit of contaminated soil to that of clean soil prepared with deionized water, is found to be a good indictor index to assess change of hydraulic conductivity of clayey soils exposed to aggressive chemical solutions (Lee et al. 2005; Du et al. 2015). Please see Page 3, Line 103 to 111.

References:

Yang, Y.-L.; Reddy, K.; Du, Y.-J.; Fan, R.-D. SHMP amended calcium bentonite for slurry trench cutoff walls: Workability and microstructure characteristics. Can. Geotech. J. 2018, 55, 528-537, doi: 10.1139/cgj-2017-0291.

Mitchell, J.K.; Soga, K. Fundamentals of soil behavior, 3rd ed.; John Wiley & Sons, Inc.: Hoboken, New Jersey, 2005.

Chapuis, R.P. Predicting the saturated hydraulic conductivity of soils: a review. B. Eng. Geol. Environ. 2012, 71, 401-434, doi: 10.1007/s10064-012-0418-7.

Du, Y.J.; Fan, R.D.; Reddy, K.R.; Liu, S.Y.; Yang, Y.L. Impacts of presence of lead contamination in clayey soil–calcium bentonite cutoff wall backfills. Appl. Clay Sci. 2015, 108, 111–122, doi: 10.1016/j.clay.2015.02.006.

Lee, J.M.; Shackelford, C.D.; Benson, C.H.; Jo, H.Y.; Edil, T.B. Correlating index properties and hydraulic conductivity of geosynthetic clay liners. J. Geotech. Geoenviron. Eng. 2005, 131, 1319-1329, doi: 10.1061/(asce)1090-0241(2005)131:11(1319).

Comment 3: The results can be impacted by many testing conditions. Please explain.

Authors’ reply: The results demonstrate that hydraulic conductivity of the bentonites is impacted by several factors including void ratio, CMC content, and metal concentration as shown in Figs. 7 to 9. It is well understood that hydraulic conductivity of untreated clayey soils including CaB and SACaB is mainly controlled by their void ratio, and higher void ratio yields higher hydraulic conductivity. It is well understood that exposure to metal-laden chemical liquids can result in increased hydraulic conductivity of bentonites. This is because of the reduced diffused double layer (DDL) thickness of bentonite, resulted from the ion exchange reactions between metal (e.g., Pb, Zn and Ca) ions in chemical liquids and exchangeable cations initially adsorbed on the bentonite particles (e.g., sodium and calcium ions). Resistance of CMC to attack of metals (e.g., Pb, Zn and Ca) is attributed due to the chemical interaction between carboxylate (COO-) groups on CMC chains and metals in the chemical liquids in the form of either unidentate, bidentate, or bridging. The resistance would increase with increased CMC content due to larger amount of carboxylate (COO-) groups on CMC chains. Consequently, restriction of increase in hydraulic conductivity of bentonites would be enhanced with CMC treatment and increased CMC content. We have addressed this in the original submission, and please see Page 12, Line 358 to 369 of the revised manuscript.

Comment 4: More details about statistical analysis in result analysis is required.

Authors’ reply: Thanks for the comment. In this study, a unique relationship between hydraulic conductivity ratio (HCR) and filtrate volume ratio (FVR) was established to quantify the change in hydraulic conductivity of both CMC-SACaB and SACaB due to exposure to heavy metals-laden water and actual landfill leachate. The regression analysis was run using a Least-Square-Root method and it generates an equation ( Eq. (7)) with a coefficient of determination (R2) of 0.883. A further statistical analysis indicates that the residual sum of squares is 3.733, and 95% confidence and prediction bands are added in Fig. 10. The 95% confidence and prediction bands reflect that if the test procedure was repeated for a large number of times, 95% of the intervals would cover the true value (Svensson et al. 2005). Please Page 13, Line 406 to 407 and Figure 10.

References:

Svensson, T.; Johannesson, P.; Maré, J.d. Fatigue life prediction based on variable amplitude tests—specific applications. Int. J. Fatigue. 2005, 27, 966-973, doi: 10.1016/j.ijfatigue.2004.11.010.

Comment 5: Error bars are required in some figures.

Authors’ reply: We did not use identical samples for evaluating liquid limit, swell index and hydraulic conductivity in this study. Therefore, error bars are not given in Figs 3, 4, 7 and 9. Three trials were conducted to determine specific gravity as per ASTM D854. The error analysis indicates that the variation of coefficient is lower than 0.46%. Please see Page 6, Line 182 to 185.

Comment 6: Please review and add the following two refences about the restriction of contaminant movement.

Bai, X., Song, K., Liu, J., Mohamed, A. K., Mou, C., & Liu, D. (2019). Health Risk Assessment of Groundwater Contaminated by Oil Pollutants Based on Numerical Modeling. International journal of environmental research and public health, 16(18), 3245.

Zhao, S., Huang, G., An, C., Wei, J., & Yao, Y. (2015). Enhancement of soil retention for phenanthrene in binary cationic gemini and nonionic surfactant mixtures: Characterizing two-step adsorption and partition processes through experimental and modeling approaches. Journal of hazardous materials, 286, 144-151.

Authors’ reply: Thanks for the comment. Results in Bai et al. (2019) are useful to under the importance of engineered barriers at contaminated sites and landfill sites. The sorption of contaminants on bentonite is one of the mechanisms to control the transport of contaminants in bentonite engineered barriers, such as soil-bentonite slurry-trench cut-off walls, geosynthetic clay liners (GCL), and compacted soil-bentonite liners. Sorption experiments presented by Zhao et al. 2015 can be used to evaluating the sorption of heavy metals on CMC-treated SACaB, which contributes to assessing effect of CMC treatment on sorption capacity of SACaB. Please see Page 1, Line 37 to 38, and Page 15, Line 455 to 457.

Comment 7: Please compare the results in this study with those in previous studies.

Authors’ reply: Thanks for the comment. Liquid limit and swell index of carboxymethyl cellulose (CMC)-treated NaB bentonite (HYPER clay) were reported in previous studies (Di Emidio et al. 2015; Malusis and Emidio 2014). In this study, variations in liquid limit and swell index and swell index ratio with CMC content in bentonite or metal concentration are compared with the results obtained from aforementioned studies. The comparison shows CMC-SACaB possesses a comparable swell index with that of HYPER clay reported by Di Emidio et al. (2015) when CMC content in Ca-bentonite was 10%. Please see 8 to 9, Line 266 to 275. In addition, CMC-SACaB in Ca solution with metal concentrations of 50 and 100 mmol/L exhibits a comparable swell index ratio with HYPER clay. However, swell index ratio of the CMC-SACaB is lower than that of HYPER clay when Ca concentration is 500 mmol/L. It is noted that as of today there are no studies on index properties of HYPER clay exposed to heavy metals-laden water and actual landfill leachate, which is the new of this study. Please see 9 to 10, Line 283 to 313.

The authors’ previous study (Liu et al. 2016) presented the results of hydraulic conductivity of sodium bentonite and sodium activated calcium bentonite exposed to DIW, Ca solution and Cr(VI) solution measured using modified fluid loss test. In this study, the hydraulic conductivity of CMC-SACaB is compared with that reported by Liu et al. (2016) at a given ranges of metal concentration and void ratio to better understand the effect of metal concentration and CMC treatment on hydraulic conductivity of bentonite. In addition, Liu et al. (2016) indicated that there existed a unique relationship between filtrate volume after 30 minutes and hydraulic conductivity obtained from the MFL tests for both clean and heavy metal- (lead, cadmium, and hexavalent chromium) contaminated sodium bentonites and sodium activated calcium bentonite under a given applied overall pressure. Based on the results, a unique relationship between filtrate volume ratio and hydraulic conductivity ratio for various types of bentonites and chemical liquids is established to quantify the change in hydraulic conductivity of CMC-SACaB and SACaB due to exposure to heavy metals-laden water and actual landfill leachate contamination. Page 13 to 14, Line 402 to 416.

References:

Di Emidio, G.; Mazzieri, F.; Verastegui-Flores, R.D.; Van Impe, W.; Bezuijen, A. Polymer-treated bentonite clay for chemical-resistant geosynthetic clay liners. Geosynth. Int. 2015, 22, 125-137, doi: 10.1680/gein.14.00036.

Malusis, M.A.; Emidio, G.D., Hydraulic conductivity of sand-bentonite backfills containing HYPER clay. Proceedings of Geo-Congress 2014: From Soil Behavior Fundamentals to Innovations in Geotechnical Engineering, Atlanta, GA, 2014; ASCE: Reston, VA.

Liu, S.Y.; Fan, R.D.; Du, Y.J.; Yang, Y.L., Modified fluid loss test to measure the hydraulic conductivity of heavy metal-contaminated bentonite filter cakes. Proceedings of Geo-Chicago 2016: Sustainable Geoenvironmental Systems, Chicago, IL, 2016; ASCE: Reston, VA.

Comment 8: The language also needs to be improved.

Authors’ reply: Thanks for the comment. The entire paper has been edited by an English professional.
